# Carob Syrup: Prebiotic Potential of a Neglected Functional Beverage of Mediterranean Countries

**DOI:** 10.3390/foods13244172

**Published:** 2024-12-23

**Authors:** Lorenzo Nissen, Davide Addazii, Flavia Casciano, Francesca Danesi, Maria Teresa Rodriguez-Estrada, Dario Mercatante, Siwar Ben Ayache, Achour Lotfi, Anagnostis Argiriou, Georgia Ayfantopoulou, Andrea Gianotti

**Affiliations:** 1DiSTAL—Department of Agricultural and Food Sciences, Alma Mater Studiorum—University of Bologna, Viale Fanin 44, 40127 Bologna, Italy; davide.addazii2@unibo.it (D.A.); francesca.danesi@unibo.it (F.D.); maria.rodriguez@unibo.it (M.T.R.-E.); dario.mercatante2@unibo.it (D.M.); andrea.gianotti@unibo.it (A.G.); 2CIRI—Interdepartmental Centre of Agri-Food Industrial Research, Alma Mater Studiorum—University of Bologna, Piazza G. Goidanich, 60, 47521 Cesena, Italy; 3BIOLIVAL—Bioresources: Biologie Integrative & Valorisation, Institut Supérieur de Biotechnologie de Monastir (ISBM), Av. Taher El Hadded BP74, Monastir 5000, Tunisia; benayache.siwar@outlook.fr (S.B.A.); lotfiachour@yahoo.fr (A.L.); 4UOA-FNS—Department of Food Science and Nutrition, University of the Aegean, University Hill, 81100 Mytilene, Greece; notis.argiriou@certh.gr; 5CERTH/HIT—Centre for Research and Technology Hellas—Hellenic Institute of Transport, 6th km Charilaou—Thermi Rd., Thermi, 57001 Thessaloniki, Greece; gea@certh.gr

**Keywords:** locustbean, 5-hydroxymethylfurfural, volatilome, multivariate analysis, bifidobacteria, lactobacilli, probiotics

## Abstract

Carob syrup, a traditional Mediterranean functional beverage obtained from *Ceratonia siliqua* (L.) pods, has been historically valued for its nutritional properties but is currently underutilized. This study compared the prebiotic potential of three handmade carob syrups produced by Tunisian women with commercial benchmarks from Italy, Greece and Cyprus. The prebiotic activity was evaluated by prebiotic scores, bifidogenic activity and volatilome characterization (SPME GC/MS) together with physicochemical and nutritional parameters. The results showed that Tunisian handmade products exhibited lower growth of pathogenic Escherichia coli compared to commercial samples. The prebiotic activity, tested against probiotic lactobacilli and bifidobacteria mixtures, showed a hierarchy of efficacy: fructo-oligosaccharides (FOSs) > Tunisian handmade products > Greek and Cypriot benchmarks > Italian benchmark. Volatilome analysis revealed about 40 compounds, mainly organic acids and aldehydes, with higher concentrations in handmade products. Positive correlations were found between prebiotic activity and short-chain fatty acids and n-hexadecanoic acid, while furfural showed negative correlations. The Tunisian artisanal products showed a higher prebiotic potential compared to the commercial counterparts, due to their higher content and diversity of organic acids. However, the presence of furfural in Tunisian products needs to be monitored due to potential toxicity concerns.

## 1. Introduction

The carob tree (*Ceratonia siliqua* L.) is an evergreen plant of the *Fabaceae* family that grows mainly in the Mediterranean region. Carob fruits consist of an outer pod with a fleshy pulp containing seeds [1]. The worldwide distribution of carob is mainly due to the multiple uses of its fruits in the food sector but also in zootechnical and pharmaceutical fields. Some traits of the carob bean make it particularly attractive to market, as the fruit is free of caffeine and theobromine, which are valuable characteristics for both human and animal nutrition, and, due to its properties, it is a viable alternative to the use of cocoa powder in the food industry [2,3]. It is widely used as a thickening and gelling agent, in particular in the plant-based food industry [4]. The rise in global production of gluten-free foods is leading to increasing demand for thickeners that can be used as technological replacers of gluten. One of them is E-410, which is derived from carob seeds and is highly requested for organic products [4]. It can be used in the baking industry as a natural flavor substitute for coffee [5]. Carob syrup is a dense, dark liquid obtained by dissolving and concentrating the fruit carbohydrates. Traditionally, the preparation process involves long extraction times and concentration by boiling the pulp in water [6]. Such long exposure to boiling temperatures leads to oxidation and degradation of some bioactive compounds in the fruit [7]. Carob syrup is characterized by a high sugar content, ranging from 60% to 80%, primarily composed of fructose, glucose and sucrose. The nutritional profile reveals minimal protein content and negligible lipid concentrations [8,9]. Of particular interest is its mineral composition, which includes macroelements (potassium, calcium, magnesium and sodium) and microelements, predominantly iron and zinc [8]. Carob syrup contains bioactive compounds that have antioxidant and antimicrobial activities in the human body [10]. These compounds are mainly represented by polyphenols, non-enzymatic browning molecules (HMF, melanoidins) and volatile substances. While several studies have investigated the presence of these compounds and evaluated their beneficial bioactivity in vitro [7,8], the prebiotic potential of carob products and the role of those bioactives remain largely unexplored. It is reasonable that, due to the presence of sugars, fibers and bioactives, the carob products can exert a prebiotic nature. Thus, the novelty of this research is characterization of the prebiotic potential of carob syrups, that so far is poorly described, and to compare it among homemade and commercial carob syrups, revealing also the volatile organic compounds (VOCs) that could be culprits of such prebiotic activity. Other innovations of this study are related to the description of prebiotic potential by untargeted omics approaches and multivariate statistical analysis. This research will contribute to the scientific understanding of carob syrup’s functional properties while supporting the preservation and promotion of sustainable, local production practices in Mediterranean communities.

## 2. Materials and Methods

### 2.1. Samples and Controls

Five samples of carob syrup were obtained from different suppliers and indicated as follows: CARGR, CARIT, CARCY, CARS1, CARS2 and CARS3. Products CARCY (Cyprus), CARGR (Greece) and CARIT (Italy) were purchased from e-commerce shops and stored at room temperature until the analyses, which were carried out long before their expiration date. Products CARS1, CARS2 and CARS3 are three Tunisian artisanal products produced in the Teboulba region, Bekalta region and Moknine region, respectively, which are included in the European PRIMA Project GourMed 2021–2024 (www.gourmed-prima.com accessed on 15 June 2024)).

### 2.2. Preparation of Carob Syrup

According to the traditional method, carob syrup was prepared in three main steps, namely: extraction, filtration and concentration. The process consists of washing and sun-drying the carob pods (pulps + seeds) until complete dehydration, followed by crushing into a traditional mortar. Six kilograms of ground pods was mixed with 15 L of water and kept overnight. The next day, the maceration was filtered through a cloth. The filtrate was then boiled for about 6 h until a syrup consistency was obtained, then conserved in glass bottles at 4 °C [11].

### 2.3. Microbial Strains and Culture Conditions

All probiotic microbial strains were obtained from commercial cultures (Bromatech, Albese, CO, Italy) or deposited type strains or probiotic-like strains belonging to the Culture Collection of DISTAL at the University of Bologna. *Escherichia coli* strains were deposited type strains. Probiotics have been previously isolated from commercial supplements and repeatedly propagated in our laboratory [12]. *Lactiplantibacillus plantarum* subsp. *plantarum* NCIMB 8299, *Lactobacillus acidophilus* LA1, *Lactobacillus rhamnosus* C1112 *Lactobacillus rhamnosus* HN001, *Lactobacillus reuterii* LR92, *Bifidobacterium bifidum* NCIMB 700795, *Bifidobacterium lactis* BL-04, *Bifidobacterium bifidum* BB-06, *Bifidobacterium breve* BB-03, *Bifidobacterium longum* BL-05, *E. coli* ATCC 25922 and *E. coli* NCIMB 555 were cultured from glycerol stocks stored at −80 °C and were propagated in selective media (Oxoid, Thermo Fisher Scientific, Waltham, MA, USA) in specific conditions [13].

### 2.4. pH, Aw Measurement and Proximate Composition of Carob Syrups

The pH was determined with a pH-meter (Crison, Alella, Spain) at 20 °C appropriately calibrated with three standard buffer solutions at pH 9.21, pH 4.00 and pH 2.00. The pH values were measured in duplicate at three different times to monitor the fermentation. Water activity was measured with the AquaLab apparatus (Decagon Devices Inc., Pullman, WA, USA), as previously described [14].

The proximate composition was determined using official AOAC methods [15] and in accordance with Regulation (EU) 1169/2011. The energy value was calculated as the sum of the energy contributions of the macronutrients.

### 2.5. Bacterial Culture-Dependent Counting

For all bacteria, 1 mL of each sample was aseptically transferred into a sterile tube with 9 mL of physiological solution (0.9 g/dL NaCl) to be serially diluted (1/10) and plated in duplicates. Lactobacilli mix was counted on MRS agar (Oxoid, Thermo Fisher Scientific, USA) after propagation for at least 24 h at 37 °C in jars with anaerobiosis catalyst (Oxoid Thermo Fisher Scientific, USA). Bifidobacteria mix was counted on MRS agar supplemented with 0.05 g/dL L-cysteine (Sigma, St. Louis, MO, USA) after propagation in the same conditions of lactobacilli. *E. coli* mix was counted on BHI agar (Oxoid, USA) after propagation at 37 °C for 24 h [13].

### 2.6. Quantification of Bacterial DNA by qPCR

Bacterial DNA from fermented hemp bran and from broths for prebiotic activity assay was extracted with the Pure Link Microbiome kit (Invitrogen, Thermo Fisher Scientific, USA). Genetic standards for qPCR were prepared from serially diluted PCR products (1/10) obtained by amplifying gene targets with specific primers (Appendix A) using the ProFlex PCR System (Thermo Fisher Scientific, USA) and SuperFi Platinum Taq (Thermo Fisher Scientific, USA) and purified with a GeneJet PCR purification kit (Thermo Fisher Scientific, USA). qPCR was performed with a QuantStudio 5 (Applied Biosystem, Foster city, CA, USA) and the QuantStudio Design and Analyse 2.1 software (Applied Biosystem, USA). PCRs and qPCRs were carried out according to previously published protocols [16].

### 2.7. Prebiotic Score

Carob syrup samples were used for calculating the prebiotic score with the related formula from two independent experiments and triplicates as previously described [17], including qPCR quantifications [16]. Carob syrups were freeze-dried using a Savant freeze-dryer Lyolab 3000 apparatus (Thermo Fisher Scientific, USA), in order to add a 1 g/dL of product to 10 mL of culture media. FOS from chicory (Sigma, USA) was used as a prebiotic positive control and 1 g/dL of glucose was used as a negative control. All bacterial mixtures were used at the final concentration of 6 Log CFU/mL [17].

### 2.8. Volatilome Analyses by Solid-Phase Microextraction–Gas Chromatography–Mass Spectrometry (SPME-GC-MS)

The volatilome was obtained from the volatile organic compound (VOC) profiles of carob syrup samples in order to identify bioactive volatile molecules that could have been implicated in the prebiotic activity. The determination of VOCs was carried out on an Intuvo Agilent 7890A gas chromatograph (Agilent Technologies, Santa Clara, CA, USA) coupled to an Agilent Technologies 5975 mass spectrometer operating in the electron impact mode (ionization voltage of 70 eV). The protocols for SPME-GC-MS analyses and for the identification of VOCs were previously published [13,16]. Briefly, before sampling, the SPME fiber was exposed to the GC inlet for 10 min for thermal desorption at 250 °C in a blank sample. 4-methyl-2-pentanol (Merck, Darmstadt, Germany) was used as an internal standard at a final concentration of 20 ppm and was left equilibrating at 40 °C for 10 min in a water bath. The SPME fiber was exposed to each sample for 40 min, and then inserted into the injection port of the GC for 10 min for sample desorption. The temperature program was: 50 °C for 1 min, then programmed at 1.5 °C/min to 65 °C, and finally at 3.5 °C/min to 220 °C, which was maintained for 25 min. Injector, interface and ion source temperatures were 250 °C, 250 °C and 230 °C, respectively. Injections were carried out in splitless mode and helium (3 mL/min) was used as carrier gas. Identification was obtained with the NIST 11 MSMS library and the NIST MS Search program 2.0 (NIST, Gaithersburg, MD, USA).

### 2.9. Prebiotic Potential

The prebiotic potential was obtained by statistical comparison of the results obtained from the microbial, the metabolic and some physiological (such as pH and A_w_) or nutritional data. This study was performed employing normalized datasets of both types of significant variables (microbial and physiological–nutritional) in a Spearman rank correlations analysis, which was then visualized in the R environment with a two-way joining heatmap that also described dendrogram hierarchy [16,18].

### 2.10. Data Processing and Statistical Analysis

All statistical analyses were performed using TIBCO Statistica 8.0 (Tibco Inc., Palo Alto, CA, USA). Normality was checked with the Shapiro–Wilk test and homoscedasticity was evaluated with Levene’s test [19]. Differences among samples were evaluated with analysis of variance (ANOVA), while principal component analysis (PCA), K-means clustering, Spearman rank correlations, two-way joining heatmap and MANOVA were used to study the relationship between the variables [13]. To compare a sample with another with the same dependent variables, a Student’s *t*-test was employed (*p* < 0.05), while to compare different cases and different variables a Tukey’s honestly significant difference (HSD) test (*p* < 0.05). was used. For descriptive statistics of the nutritional data, mean values, ranges (minimum–maximum) and coefficient of variation (CV% = standard deviation/mean × 100) were calculated to characterize the variability of each parameter across samples. For PCA and Spearman rank correlations, the dataset was normalized using the mean centering method. All results are expressed as mean values obtained at least from duplicate batches in two independent experiments. qPCR and pH results were obtained from three replicates from two independent experiments.

## 3. Results and Discussion

### 3.1. pH, A_w_ and 5-Hydromethylfurfural (5-HMF) Values of Carob Syrups

Table 1 reports the pH, Aw and 5-HMF content of the carob syrup samples, which significantly differed within the syrup types, as well as between homemade and commercial ones. The average pH of syrups was 4.87 ± 0.10; CARIT and CARGR had the highest pH values, while the Tunisian products displayed the lowest ones, but all of the syrup samples agree with data from the literature [20]. Mean Aw of samples was 0.81 ± 0.02, displaying significant differences within and between both types of products, even though the values are all within the expected range [20]. In this case, the concentration conditions and technologies used for the syrup production might have contributed to diverse concentration levels of the syrup. Regarding 5-HMF content, a large variability was found, especially in home-made syrups where CARS1 had a much higher concentration of this compound. This parameter also varied among commercial products, CARCY being the product with the lowest 5-HMF content. 5-HMF is generated in carob syrup mainly by dehydration of monosaccharides during caramelization; it can also form, to a lesser extent, by thermal decomposition of reducing sugars (fructose and glucose) in the presence of amino compounds by means of a Maillard reaction. In this study, the types and concentrations of carbohydrates in the raw materials, together with the processing conditions (temperature/time) used for the home-made syrups, might have largely contributed to such variability. Considering that 5-HMF has shown adverse health and toxicity effects on various animal experiments [21], regulatory limits about its content in food have been defined [21]. Moreover, to lower 5-HMF level in fruit syrups, the reduction of the heating temperature during syrup concentration has proved to be a successful strategy [22].

The proximate analysis of carob syrups revealed distinct nutritional profiles between homemade and commercial products (Table 2), reflecting differences in both raw materials and processing technologies. Moisture content was found to be a critical parameter influencing the overall nutrient density of the products. Commercial syrups from different Italian, Greek and Cypriot producers showed greater variability in water content compared to homemade Tunisian products, with CARIT having significantly higher moisture, while CARGR and CARCY had lower values. This variation suggests different technological approaches to water removal during processing at an industrial level [23]. The homemade Tunisian samples had more consistent intermediate moisture levels, indicating a more uniform traditional processing approach, despite the lack of industrial dehydration equipment. As expected from the natural composition of carob, sugars were the predominant nutrient in all samples, confirming that carob pods are an excellent source of natural sweeteners [23,24]. This characteristic makes carob syrup a potential natural alternative to conventional sweeteners, which is particularly relevant in the current context of growing demand for natural sugar alternatives [25]. The carbohydrate content, which is mainly sugar as expected, showed an inverse relationship with the moisture content. The Greek and Cypriot commercial products achieved higher carbohydrate concentration through more efficient dehydration processes, whereas the Italian product retained more moisture, resulting in lower carbohydrate density. The homemade products maintained intermediate carbohydrate levels, suggesting that traditional processing methods achieve a reasonable but less extreme concentration than industrial processes. An interesting finding emerged from the protein profiles, where homemade products consistently showed higher protein retention compared to commercial variants. This difference suggests that industrial processing methods may affect protein retention differently than traditional techniques, possibly due to thermal processing conditions and/or preliminary clarification steps that may remove protein-rich components. From a technology perspective, these differences in nutritional composition reflect the impact of different processing approaches. Industrial methods appear to achieve more extreme moisture removal, resulting in higher sugar concentration [26], presumably to ensure a longer shelf-life. Traditional homemade processing, although less standardized, appears to achieve moderate levels of concentration. These findings provide valuable insights for optimizing carob syrup production, suggesting that different processing parameters could be adjusted to achieve desired nutritional profiles and shelf-life.

### 3.2. Bacterial Enumeration by Growth on Selective Media and qPCR

Microbial selective growth in the presence of carob syrups with enumerations obtained from direct plate counting and from qPCR was independently tested against three different mixes of bacteria, i.e., the mix of lactobacilli, the mix of bifidobacteria and the mix of *E. coli* (Figure 1A). Specific bacterial growth was performed and compared on specific culture media with the addition of 10% (*w/v*) of each carob syrup samples from the Tunisian home-made products (CARS1, CARS2 and CARS3) and with the addition of 10% (*w/v*) of each of the benchmark products from three Mediterranean countries: Italy (CARIT), Greece (CARGR) and Cyprus (CARCY). Considering the growth of lactobacilli mix (Figure 1B), the addition of 10% *w/v* to the culture medium of benchmark products from Greece (9.43 ± 1.99 Log cells/mL) and from Cyprus (9.86 ± 1.25 Log cells/mL) showed similar results to that determined by the addition of the positive control FOS (9.87 ± 1.13 Log cells/mL). Among the three homemade products, only CARS2 (9.43 ± 1.01 Log cells/mL) gave similar results to those of the Greek and Cypriot benchmarks, while CARS1 (8.66 ± 1.47 Log cells/mL) and CARS3 (8.60 ± 1.16 Log cells/mL) were similar to CARIT (9.06 ± 1.09 Log cells/mL), but different from the positive control FOS. Interestingly, the growth of the bifidobacteria mix was larger for CARS3 than for any homemade products or any benchmark products. CARS3 (9.49 ± 2.40 Log cells/mL) had similar values to those of FOS (9.61 ± 2.64 Log cells/mL) (Figure 1C). Considering the selective growth of the *E. coli* mix, CARS2 was the sample that made the mix grow more (9.31 ± 2.82 Log cells/mL), but with values similar to the positive and negative controls, meaning that it has the same intensity in counteracting the growth of *E. coli* mix. Interestingly, the other two homemade products were able to limit the growth of *E. coli* mix with more intensity, as their values were 8.92 ± 1.04 Log cells/mL and 9.32 ± 1.55 Log cells/mL for CARS1 and CARS3, respectively.

### 3.3. Prebiotic Score

The prebiotic score of FOS towards lactobacilli mix was 0.36 ± 0.11, similar to values reported in the literature (Figure 2A) [17]. Carob syrup had prebiotic activity towards the lactobacilli mix to differing extents depending on the product. In particular, the homemade product CARS2 had a prebiotic activity similar to that of the positive control FOS and two benchmarks, CARGR and CARCY, and a higher prebiotic activity with respect to the Italian benchmark CARIT. CARS2 had a similar prebiotic activity to CARIT, while CARS3 had no prebiotic activity towards lactobacilli. All samples tested had prebiotic activity towards bifidobacteria. In particular, the prebiotic score of homemade product CARS1 was the highest, being similar to the Greek benchmark product CARGR. Interestingly, this score was higher than that of FOS and of the Italian and Cypriot benchmarks (Figure 2B).

#### 3.3.1. VOCs Related to Prebiotic Activity

To investigate in depth the effective prebiotic activity of carob syrups, their volatilome was characterized by SPME-GC-MS to identify those molecules related to prebiotic activity. Considering the general scenario, the sample with the richest volatilome in terms of chemical speciation of metabolites was the Tunisian one from Teboulba CARS1, with 71 different VOCs (Figure 3A). Among the benchmark products, the Italian benchmark CARIT had the highest number of VOCs (70), followed by CARS2 from the Bekalta region. Seventeen VOCs were shared among all samples (Figure 3B) which could be considered as the signature of carob syrups: (-)-epinephrine, acetic acid, benzoic acid, butanoic acid, butylated hydroxytoluene, eicosane, ethanone,1-(1H-pyrrol-2-yl)-, 5-hydroxymethylfurfural, hexadecane, octadecane, 2,4-bis(1,1-dimethylethyl)-phenol, 2-chloro-4-(1,1-dimethylpropyl)-phenol, 4-(1,1,3,3-tetramethylbutyl)-phenol, 2-methyl-propanoic acid and tetradecane (Appendix A). It should be pointed out that the three Tunisian carob syrups had exclusive and characterizing VOCs, e.g., acetaldehyde for CARS1; 6-chloro-2-methylquinolone and dimethyl trisulfide for CARS2; 2,3-dihydro-1H-inden-5-ol for CARS3.

#### 3.3.2. Volatilome Analysis of Aldehydes, Ketones, Alcohols and Phenolic Compounds

The complete volatilome of every tested sample was composed of 88 VOCs and the sample CARIT had the highest concentration of most of the VOCs, followed by CARS1 and CARS2 (Figure 4A and Appendix A). VOCs were classified according to their chemical classes and samples were compared focusing on ANOVA significant variables (*p* < 0.05) and the single chemical class. Therefore, the volatilome was studied in four datasets of VOCs, i.e., aromatic alcohols, organic acids, aldehydes and ketones, amines and similar. These datasets were submitted to a multivariate approach, including targeted PCA and targeted MANOVA (*p* < 0.05). Considering aromatic alcohols (Figure 4B), 12 compounds were able to discriminate the carob syrup samples. In particular, the three Tunisian samples were characterized by specific VOCs. CARS3 was characterized by two bioactive phenol derivates and one indole derivate; among these compounds, 2,3-dihydro-1H-Inden-5-ol is the most interesting, as it has antimicrobial and antioxidant activity and a pleasant odor described as green, woody, cedar pine and licorice [27]. CARS1 was characterized by another derivate of indole, i.e., 5-methyl-2-phenyl-1H-indole, which possesses antioxidant and antiinflammation activity [28] and antimicrobial activity versus pathogenic *Candida albicans* [29]; orcinol was also a characterizing VOC of CARS1, which is a bioactive molecule that is able to inhibit melanogenesis and induce apoptosis on cancer cell lines [30] and in its ether form is the principal descriptor of the scent of rose petals [31]. Boubleza and colleagues described that carob pods are rich in natural antioxidants similar to synthetic hydroxytoluene, such as orcinol [32]. Considering organic acids, most organic acids characterized the benchmarks, but some of them were descriptors of CARS2, such as *n*-hexadecanoic acid, *n*-decanoic acid and citric acid (Figure 4C). The first two organic acids are medium-chain organic acids that are generated by oxidation of fatty acids or of the corresponding aldehydes; they display a renowned bioactivity in humans, including strengthening of the epithelial barrier function of the gut and potentiating immunity [33]. Atta and colleagues reported that, in carob pod extract, *n*-hexadecanoic acid was the third in abundance among all compounds found by GC/MS characterization [34]. Citric acid could have an added advantage in the composition of a beverage, as it is a potent food preservation agent, acting as an acidifier and chelating agent [35]. The presence of organic acids represents a prebiotic stimulus as a good fermentative substrate for lactic acid bacteria [36]. Considering aldehydes and ketones, PCA was able to discriminate the carob syrup samples. 1-hydroxy-2-butanone (also known as acetoin), a sugar degradation product, is a descriptor of CARS2 that confers a buttery aroma (Figure 4D). This latter compound is reported in the literature to be a result of microbial fermentation of sugar-rich plant-based syrups [37]. In carob syrups, aldehydes are mainly generated by direct degradation of the sugar carbon chain (after isomerization and/or dehydration) and also by condensation of the fragments [38]. Certain aldehydes are health promoters because they contribute positively to cell homeostasis and microbiota eubiosis, such as indole-3-aldehyde [39], while most are detrimental, being cytotoxic at a low threshold, such as acetaldehyde [40]. Carob syrup is also rich in ketones, some of which are desirable, such as acetoin, while others (such as furanones) are unwanted, as they could be toxic for the host, acting, for example, as quorum-sensing quenchers [41]; however, it should be pointed out that several furanones provide an intensive caramel flavor [38] due to the planar vicinal enol-oxo configuration in their structure [42]. For this dataset, it is important to stress the fact that CARS1 was characterized by the presence of 5-HMF and furfural, the latter being a thermodynamically stable degradation product of 5-HMF [43]. As described in the previous section, CARS1 had the highest quantity of 5-HMF, which is an aspect that should be further investigated for safety reasons, but also furfural has cytotoxic effects already at a concentration of 10 mmol/L [44] so this should be taken into account. Considering other compounds, bioactive catabolites ephedrine-like alkaloids were descriptors of CARS2 (meta-hydroxynorephedrine) and CARS3 ((-)-Norephedrine), while bioactive 8-epi-gamma-eudesmol was descriptor of CARS1 (Figure 4E). The ephedrine derivate compounds are natural like alkaloids of plant origin [45]. Ephedra herb alkaloid has anti-inflammatory, analgesic and anti-influenza effects [46].

### 3.4. Metabolite Correlations over Prebiotic Activity

According to the Spearman rank correlation analysis (*p* < 0.05) (Figure 5), orcinol, benzenepropanal 4-(1,1-dimethylethyl) and citric acid seem to be the metabolites that are more directly related to the prebiotic activity, especially in the Tunisian syrups towards the bifidobacteria mix growth. In fact, these molecules were exclusive markers of homemade products and were not detected in the benchmark products, except for benzenepropanal 4-(1,1-dimethylethyl) which was also found in CARCY (Appendix A). Hexanoic and benzoic acids are correlated to potential prebiotic activity of beneficial lactobacilli, but not of bifidobacteria (Figure 5); these molecules were present in the homemade products. Hexanoic acid is an important block of sugar breakdown by lactobacilli and also a good substrate for further lactobacilli fermentation and an additional boost for beneficial bacteria of the colon lumen and an antifungal agent [47]. This compound is antibacterial for many Gram-negative and opportunistic intestinal microbes, such as *Escherichia coli* [48], but not for lactobacilli that use this acid as an elongation substrate to build longer chain acids [49]. Also, benzoic acid is a typical signature of lactobacilli fermentation and has antifungal activity [50] and protective activity versus Gram-negative *Enterobacteriaceae* in the intestinal tract [51]. In general, the carob syrup products showed a variable VOC profile, which could be due to both the raw material quality and the different preparation/cooking methods. For this type of product, it is important to establish standard preparation conditions to try to obtain benchmark-like products; in this way, it will be easier to achieve the desired product specifications and meet consumer preferences or specific dietary requirements.

## 4. Conclusions

In this work we studied the prebiotic activity and potential of carob syrups, comparing benchmark products with a large distribution to regional homemade carob syrups obtained by a Tunisian partner of the PRIMA GourMed project. The definition of a prebiotic is “a substrate that is selectively utilized by host microorganisms conferring a health benefit” [52]. Therefore, we adopted a methodology for prebiotic activity and prebiotic potential that is based on the growth of three different bacterial groups inhabiting the human gut ecosystem: (i) probiotic bifidobacteria, (ii) probiotic lactobacilli and (iii) enteropathogenic *E. coli* strains and on the volatile metabolites of the substrates. The artisanal Tunisian homemade products seem to have prebiotic potential both towards lactobacilli and bifidobacteria, but to different extents. Such differences can be attributed to the raw materials, formulation and process leading to a higher content and a greater variety of organic acids, that could have fostered more the probiotic growth. 5-HMF was also involved in the prebiotic potential of Tunisian homemade products, but its content should be studied further. This feature could be ascribed to an excessive duration of cooking or to uncontrolled cooking temperatures, as 5-HMF is mainly generated in carob syrup by dehydration of monosaccharides during caramelization and, to a lesser extent, by Maillard reactions. To conclude, the homemade Tunisian products had a higher or comparable prebiotic activity to the benchmark products. However, the traditional cooking process must be standardized by properly using time/temperature parameters. From the volatilome profile, those homemade products had more abundant compounds derived from Maillard reactions than the benchmark products, therefore it seems reasonable that the traditional cooking process can be improved by lowering the temperature and shortening the time of cooking. These preliminary findings open great opportunities in the market of functional foods to homemade Tunisian products. The limits of the proposed research work are principally due to the rural recipe and folkloristic cooking process that were applied to produce the Tunisian carob syrups. In particular, these issues did not permit us to profoundly characterize the nutritional profile of the products nor monitor/control the conditions of the process. Additionally, more information should be collected on the functional impact of carob syrups on consumers, and the prebiotic signature needs more investigation. In this context, future perspectives for this work are to enlarge the knowledge on carob syrup functionality on intestinal human microbiota with an in vitro intestinal model. Notwithstanding, this work for the first time compared homemade and industrial recipes of carob syrups for their prebiotic activity and revealed the prebiotic potential that is based on the correlations among bacteria and metabolites that determine prebiotic activity.

## Figures and Tables

**Figure 1 foods-13-04172-f001:**
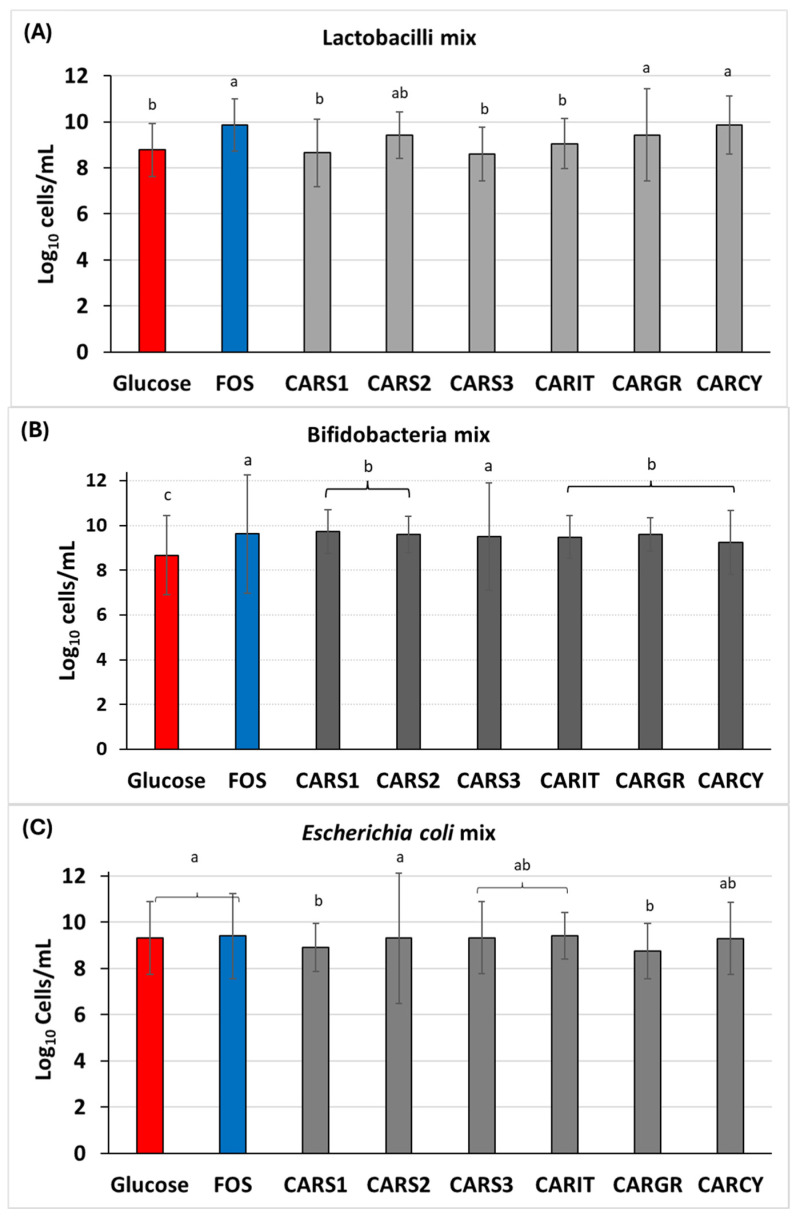
Selective bacterial growth on carob syrup handmade product and carob syrup benchmarks. (**A**) Aerobic growth on MRS agar of lactobacilli mix; (**B**) Anaerobic growth on MRS cysteine agar of bifidobacteria mix; (**C**) Aerobic growth on BHI agar of *E. coli* mix. Glucose = negative control; Fructooligosaccharide (FOS) = positive control. CARS1 = Homemade Tunisian carob syrup (Teboulba); CARS2 = Homemade Tunisian carob syrup (Bekalta); CARS3 = Homemade Tunisian carob syrup (Moknine); CARIT = Italian carob syrup (large distribution); CARGR = Greek carob syrup (large distribution); CARCY = Cypriot carob syrup benchmark (large distribution). Different letters indicate significant differences between samples by statistical analyses, as ANOVA followed by Tukey’s post hoc test (*p* < 0.05).

**Figure 2 foods-13-04172-f002:**
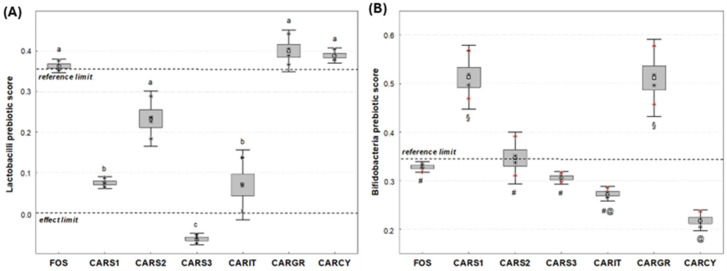
Prebiotic scores towards probiotic strains of (**A**) lactobacilli and (**B**) bifidobacteria. CARS1 = Homemade Tunisian carob syrup (Teboulba); CARS2 = Homemade Tunisian carob syrup (Bekalta); CARS3 = Homemade Tunisian carob syrup (Moknine); CARIT = Italian carob syrup (large distribution); CARGR = Greek carob syrup (large distribution); CARCY = Cypriot carob syrup benchmark (large distribution); Fructooligosaccharide (FOS) = positive control. Different letters or symbols indicate significant difference between samples by statistical analyses, as ANOVA followed by Tukey’s post hoc test (*p <* 0.05). Marker indicates the mean value, Box indicates mean and Standard Deviation, Whiskers indicate confidence interval at 95%.

**Figure 3 foods-13-04172-f003:**
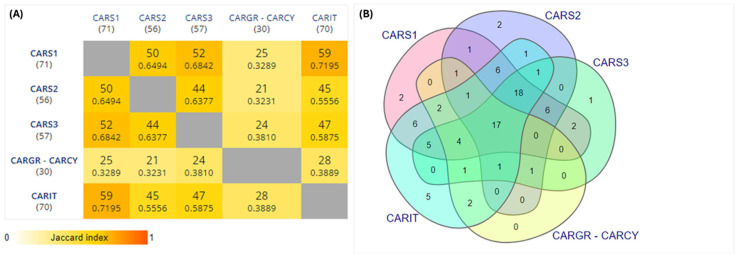
(**A**) Pairwise Intersection Map and (**B**) VENN Diagram for the volatilome of the controls and homemade products. CARS1 = Homemade Tunisian carob syrup (Teboulba); CARS2 = Homemade Tunisian carob syrup (Bekalta); CARS3 = Homemade Tunisian carob syrup (Moknine); CARIT = Italian carob syrup (large distribution); CARGR = Greek carob syrup (large distribution); CARCY = Cypriot carob syrup benchmark (large distribution). More information can be found as Appendix A. Colors of sets are specific for each sample.

**Figure 4 foods-13-04172-f004:**
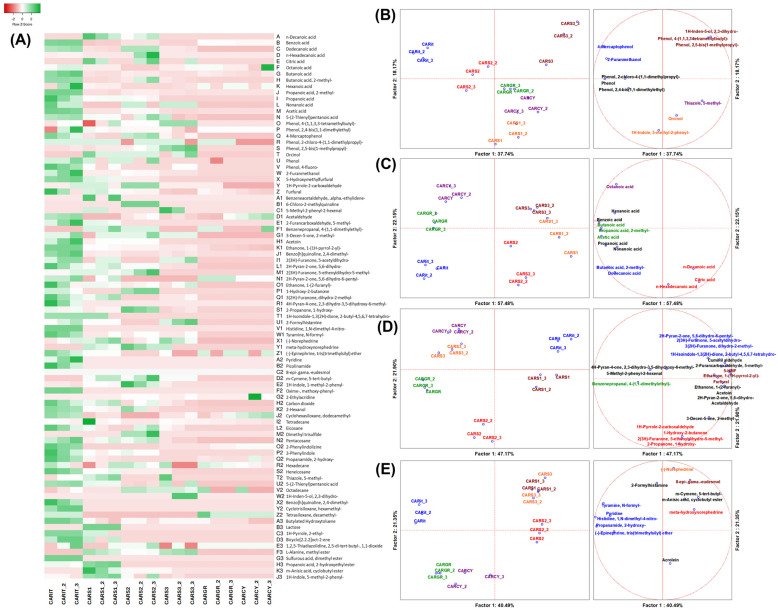
(**A**) Heatmap of volatile organic compounds (VOCs) and (**B**–**E**) PCAs of class of VOCs. (**B**–**E**) Left-side diagrams are for PCAs of cases, while right-side diagrams are for PCAs of dependent variables; variables with the same color of cases indicate specific descriptors by ANOVA; (**A**) complete volatilome with full description of VOC identity reported also in Appendix A; (**B**) PCA of aromatic alcohols; (**C**) PCA of organic acids; (**D**) PCA of aldehydes and ketones; (**E**) PCA of amines and similar VOCs. CARS1 = Homemade Tunisian carob syrup (Teboulba); CARS2 = Homemade Tunisian carob syrup (Bekalta); CARS3 = Homemade Tunisian carob syrup (Moknine); CARIT = Italian carob syrup (large distribution); CARGR = Greek carob syrup (large distribution); CARCY = Cypriot carob syrup benchmark (large distribution).

**Figure 5 foods-13-04172-f005:**
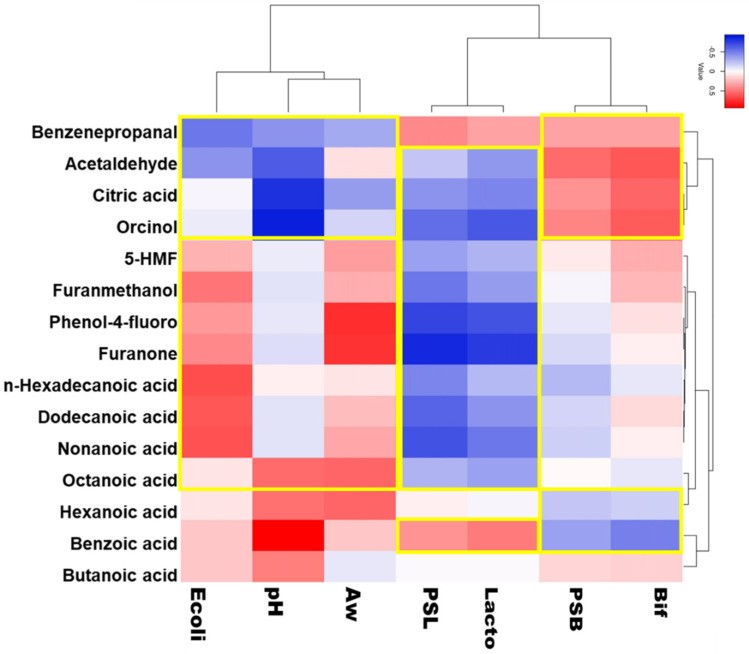
Spearman Rank Correlation Analysis. Correlations of prebiotic activity, pH and Aw values and volatile organic compounds of carob syrup products. 5-HMF = 5-hydroxymethylfurfural; Ecoli = selective growth of *E. coli* mix; PSL = prebiotic activity of lactobacilli mix; Lacto = selective growth of lactobacilli mix; PSB = prebiotic activity of bifidobacteria mix; Bif = selective growth of bifidobacteria mix. Benzenepropanal = benzenepropanal 4-(1,1-dimethylethyl). Significance of correlations by Spearman Rank analysis (*p* < 0.05) is reported in Appendix A. Yellow line indicates clusters of correlations.

**Table 1 foods-13-04172-t001:** pH, A_w_ and 5-HMF values of carob syrup samples.

Sample	Origin/Supplier	pH	A_w_	5-HMF (mg/kg)
Homemade				
CARS1	Tunisia, Teboulba	4.76 ± 0.03 ^y^	0.784 ± 0.001 *	1031.63 ± 202.63 ^a^
CARS2	Tunisia, Bekalta	4.74 ± 0.03 ^y^	0.825 ± 0.008 **	325.73 ± 33.77 ^b^
CARS3	Tunisia, Mokikne	4.90 ± 0.03 ^z^	0.824 ± 0.007 **	247.37 ± 78.48 ^c^
Commercial products				
CARIT	Italy	4.97 ± 0.01 ^z^	0.791 ± 0.001 *	723.96 ± 202.63 ^a^
CARGR	Greece	4.95 ± 0.02 ^z^	0.846 ± 0.007 ***	868.97 ± 181.13 ^a^
CARCY	Cyprus	4.82 ± 0.02 ^yz^	0.821 ± 0.003 **	329.98 ± 93.91 ^c^

Different letters or symbols within a column indicate the significant differences reported from comparisons by unpaired *t*-test (*p* < 0.05).

**Table 2 foods-13-04172-t002:** Nutrient composition of carob syrups, per 100 g.

	Homemade Carob Syrups			Commercial Carob Syrups			
Energy/Nutrient	CARS1	CARS2	CARS3	CARIT	CARGR ^†^	CARCY ^†^	Mean (Range) CV%
Energy (kcal)	263.00	251.00	256.00	233.00	285.00	286.00	262.33 (233–286) 7.14
Protein (g)	2.91	2.26	2.80	2.33	2.30	1.50	2.35 (1.50–2.91) 19.42
Carbohydrates (g)	62.72	60.28	60.99	55.74	69.20	70.00	63.16 (55.74–70.00) 7.96
Sugars, total (g)	60.30	58.70	58.00	53.60	50.00	70.00	58.43 (50.00–70.00) 10.64
Fat (g)	0.10	0.11	0.10	0.12	<0.50	<0.10	0.10 (0.10–0.12) 7.27
Water (g)	30.74	34.10	32.98	38.90	26.80	27.20	31.79 (26.80–38.90) 13.12

^†^ Values were obtained from manufacturer’s nutritional labels. CV: coefficient of variation. Data are reported as mean ± standard deviation (SD). The significant differences were reported from comparisons by unpaired *t*-test with *p* < 0.05. CARS1 = Homemade Tunisian carob syrup (Teboulba); CARS2 = Homemade Tunisian carob syrup (Bekalta); CARS3 = Homemade Tunisian carob syrup (Moknine); CARIT = Italian carob syrup (large distribution); CARGR = Greek carob syrup (large distribution); CARCY = Cypriot carob syrup benchmark (large distribution).

## Data Availability

The original contributions presented in the study are included in the article/Appendix A, further inquiries can be directed to the corresponding author.

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
