# Peer review of "Carob Syrup: Prebiotic Potential of a Neglected Functional Beverage of Mediterranean Countries"

_foods, 2024, doi:10.3390/foods13244172_

Round 1
Reviewer 1 Report
Comments and Suggestions for Authors
This manuscript has no enough quality to be published. It is not well organized, and the writing of this manuscript need be further improved. Major revision is required.
Other comments:
1. "Introduction" is too short, and the novelty of this work is not well highlighted.
2. In the text body, there are two Figure 1 and two Figure 2.
3. Between line 388-398, the information in Figure 1 is hard to read.
4. The title " 3.1. pH, Aw values, 5-hydromethylfurfural (5-HMF) and proximate composition of carob syrups" need be changed into an appropriate one.
5. "3.2.2. VOCs related to prebiotic activity" need be well summarized. How to confirm prebiotic activity? The detailed results need be described and discussed.
6. In " 3.4. Metabolites correlations over prebiotic activity", it lacks of detailed description, and in-depth discussion should be made comparing their findings with related works.
7. The advantage and disadvantage of this work need be well summarized, and the novelty of this research need be highlighted before "Conclusion".
Author Response
This manuscript has no enough quality to be published. It is not well organized, and the writing of this manuscript need be further improved. Major revision is required.
We thank Rev#1 for the valuable comments and suggestions. We have revised the paper accordingly and we hope that the new version will satisfy all.
Other comments:
- "Introduction" is too short, and the novelty of this work is not well highlighted.
We thank Rev#1 for the comments. We have revised the introduction section highlighting the novelty of the work.
- In the text body, there are two Figure 1 and two Figure 2.
We are sorry for the inconvenience; the figures’ numbers were revised, and the order now is correct.
- Between line 388-398, the information in Figure 1 is hard to read.
We are sorry, but maybe Rev#1 meant lines 288-298. Anyhow, in the revised version we have reported the citation to figure 1 at paragraph 3..2
- The title " 3.1. pH, Aw values, 5-hydromethylfurfural (5-HMF) and proximate composition of carob syrups" need be changed into an appropriate one.
We thank the reviewer for the comment, and we have revised the title as follows: 3.1. pH, Aw and 5-hydromethylfurfural (5-HMF) values of carob syrups.
- "3.2.2. VOCs related to prebiotic activity" need be well summarized. How to confirm prebiotic activity? The detailed results need be described and discussed.
We thank Rev#1 and we have revised the paragraph 3.2.2 according to the comments. The new version includes more comparison with other works related to carob products. Also, more references were added.
- In " 3.4. Metabolites correlations over prebiotic activity", it lacks of detailed description, and in-depth discussion should be made comparing their findings with related works.
We thank Rev#1 for the comment; paragraph 3.4 has been revised according to rev #1 indications and additional references were added.
- The advantage and disadvantage of this work need be well summarized, and the novelty of this research need be highlighted before "Conclusion".
We thank Rev#1 for the comments; the conclusion section of the manuscript has been revised according to rev #1 indications.
Reviewer 2 Report
Comments and Suggestions for Authors
Many errors in References
References 41-43 ? : not on the list
Reference 1 : don’t describe carob fruits
Is the same with next references 2,3, and following
References nr 12 , presented in M&M as “ in our laboratory” was published by others authors
I stopped to verified, because ..
IT SEEM THAT THE REFERENCES LIST WAS NOT FOR THAT PAPER !!! OR REFERENCES WERE MIXED
So, the paper must be rejected.
Author Response
Many errors in References
References 41-43 ? : not on the list
Reference 1 : don’t describe carob fruits
Is the same with next references 2,3, and following
References nr 12 , presented in M&M as “ in our laboratory” was published by others authors
I stopped to verified, because ..
IT SEEM THAT THE REFERENCES LIST WAS NOT FOR THAT PAPER !!! OR REFERENCES WERE MIXED
So, the paper must be rejected.
We are sorry for the inconvenience, in the revised version we have included the correct References section, that is also updated due to other REV's comments.
Reviewer 3 Report
Comments and Suggestions for Authors
Dear Authors,
I appreciate the clarity and coherence of your paper.
However, I consider that you didn’t explain, clearly enough, the practical purpose of your research.
At the same time, Conclusions are evasive. You mentioned that “However, the traditional cooking process must be standardized by properly using time/temperature parameters”. Would it be possible that you suggest limits or suggestions for these parameters?
There are some editing errors.
The reference list contains 40 works, while in the paper there are 42 references.
Author Response
I appreciate the clarity and coherence of your paper.
However, I consider that you didn’t explain, clearly enough, the practical purpose of your research.
-------We thank Rev#3 for the valuable comments and suggestions. We have improved the article in the revised version. The introduction section now includes the practical porpouse of the research
At the same time, Conclusions are evasive. You mentioned that “However, the traditional cooking process must be standardized by properly using time/temperature parameters”. Would it be possible that you suggest limits or suggestions for these parameters?
--------We thank Rev#3 for the comments. We have improved conclusion section in the revised version, giving indications to the aforementioned issue.
There are some editing errors.
---We have tried to address and correct all the errors
The reference list contains 40 works, while in the paper there are 42 references
----We have revised the reference and no errors should be present in the revised version.
Round 2
Reviewer 1 Report
Comments and Suggestions for Authors
This revised version can be accepted as it is.